# Characterization of the Tensile Properties and Nanoscale Phase Structures of Modified Asphalts and Their Aging Behavior

**DOI:** 10.3390/polym16223121

**Published:** 2024-11-07

**Authors:** Yuhui Zhang, Ming Wang, Chengwei Xing, Lingyun Zou, Jingxuan Guo, Yueduo Wang

**Affiliations:** 1School of Traffic Science and Engineering, Civil Aviation University of China, Tianjin 300300, China; caucwm@yeah.net (Y.Z.); m_wang@cauc.edu.cn (M.W.); z970924149@163.com (L.Z.); 2Key Laboratory for Intelligent Construction and Maintenance of Civil Aviation Airport, Chang’an University, Xi’an 710064, China; 3China Communications Highway Planning and Design Institute Co., Ltd., Beijing 100010, China; yueduo.wang@foxmail.com

**Keywords:** modified asphalt, force ductility, tensile properties, AFM, nanoscale phase structure, aging

## Abstract

This study focuses on exploring tensile properties and nanoscale phase structures of different modified asphalts, and their aging behavior. For this, one virgin asphalt and three modified asphalts, namely, 4% SBS-modified asphalt, 2% SBS and 20% crumb rubber (CR) composite asphalt, and 4% SBS and 2%TiO_2_ composite asphalt, were prepared and investigated using the force-ductility test and atomic force microscopy (AFM). Also, detailed experiments of short-term (STA) and long-term (LTA) aging were conducted to obtain aged asphalt specimens. The results showed that a ductile fracture was found for the three modified asphalts. However, for the activation energy, SBS asphalt and SBS&TiO_2_ asphalt were 2.87 times and 3.31 times that of SBS&CR asphalt, respectively. This demonstrates that the activation of the SBS polymer phase requires more energy during the stretching process when the rubber powder is not present. SBS&CR asphalt and SBS&TiO_2_ asphalt showed better tensile properties and aging resistance in terms of the quantitative results of tensile property indicators, indicated by a larger value of fracture ductility, tensile compliance, and the toughness ratio under the same aging condition. According to the AFM results, SBS modifier had little effect on the phase structure of virgin asphalt, while TiO_2_ modifier increased the number of bee phases and made their distribution more uniform, indicating the formation of a more stable phase structure system. This may contribute to its better tensile properties and aging resistance. Moreover, TiO_2_ molecules inhibited the aggregation behavior of polar molecules during the aging process, which led to a reduction in surface roughness. By comparison, the effect of aging on the phase structure of SBS&CR asphalt was more significant among the three modified asphalts. This result can be attributed to the interaction between rubber powder particles and asphalt.

## 1. Introduction

Modified asphalt is widely used in pavement construction because of its excellent high-temperature, low-temperature, fatigue and anti-aging properties [1,2]. In addition to common polymer-modified asphalt, nanoparticle-modified and composite-modified asphalts have also been gradually developed and tried for practical application. However, due to the influence of various factors, crack disease of pavements is common, which greatly reduces the service quality and service life of pavements. Many studies have shown that the tensile cohesion of asphalt (which determines the amount of energy required to form cracks inside the material) is closely related to the cracking behavior of the pavement [3,4]. Thus, it is of great significance to fully understand the tensile cohesion properties of modified asphalt to improve the anti-cracking ability of asphalt pavement.

However, limited by the test means, the tensile cohesion of asphalt has not been directly evaluated. Researchers often use ductility at different temperature domains (especially lower temperature conditions) to indirectly characterize asphalt cohesion strength. In the ductility test, the low-temperature deformation value at tensile fracture is generally used to evaluate the ductility of asphalt. It seems to provide very limited information regarding this stretching process. Early research results also indicate that the ductility of asphalt is indeed directly related to pavement performance [5]. Kandhal showed that pavements with large ductile asphalt have less damage, compared to other pavements with the same penetration grade [6]. With the increasing maturity of modified asphalt technology, it has been found that the performance difference of modified asphalt with the same ductility is still significant. This phenomenon is attributed to the fact that the rheological properties of modified asphalt, especially polymer-modified asphalt, are extremely complex. And the tensile cohesion cannot be reasonably predicted based on the ductility value at fracture. Anderson et al. proposed the force-ductility test as an improved test method [7]. During the test measurement, the tensile force and its corresponding deformation value can be recorded simultaneously. The force-ductility curve of the whole test process can be obtained, and multiple indexes can be calculated by means of mathematical statistics. The emergence of this method realizes the idea of evaluating asphalt cohesion from multiple dimensions. In 2010, the European specifications for polymer-modified bitumen (EN 14023 [8]) proposed a ductility test (EN 13589 [9]) to set the performance requirements in terms of cohesion. Since then, more researchers have tried to use force-ductility tests to evaluate the tensile cohesion of asphalt. A lot of work has been performed both on the characterization of the force-ductility curves of modified asphalt and on the quantitative characterization of tensile cohesion properties [10,11,12]. And results show that the force-ductility test is a very effective method to evaluate the tensile properties of asphalt.

Generally, three failure modes, namely, brittle fracture, flow failure, and ductile fracture, are recognized for different types of asphalt, based on the characteristics of the force-ductility curve [13]. Usually, base asphalt shows a flow failure, whereas most polymer-modified asphalts show a ductile fracture. With this test, the tensile performance of asphalt was evaluated by some quantitative indicators, like peak force, stretched elongation, and deformation energy. Chen et al. showed that the deformation energy obtained from the force-ductility test can be employed to evaluate the low-temperature performance of modified asphalt [14]. Bai et al. found that there was a good correlation between the ductility value at 5 °C obtained from the force-ductility test and the creep stiffness value obtained from the bending beam rheometer test. Also, it is believed that the ductility value at 5 °C is an ideal indicator for evaluating the low-temperature performance of SBS-modified asphalt [15]. In recent years, in order to better understand the characteristics of polymer-modified asphalt, some new parameters have been gradually adopted. Zhang et al. pointed out that true fracture stress was a promising indicator for modified asphalt. In addition, an “8”-shaped mold was recommended for the force-ductility test as a result of its better discrimination ability and less variability [16]. Rajan et al. utilized an additional parameter, work performed for polymer activation (WPA) associated with the force-ductility curve. Also, WPA was assumed to be an important parameter for understanding the transition phase of polymer-modified binders [17]. In addition, the normalized post-peak deformation energy was proposed for the ductility test analysis [13]. However, Qian reported that the conclusion from the two indicators, namely, ultimate elongation and deformation energy, seem to be inconsistent according to the test results of crumb rubber/SBS-modified asphalt. Hence, it is recommended to evaluate the low-temperature properties of modified asphalt with multiple indicators [18]. In a word, the researchers conducted a more in-depth discussion and analysis on the tensile cohesion characteristics of modified asphalt, and have given many suggestions on the applicability of evaluation indicators. However, there are few studies on the evolution behavior of tensile cohesion of different types of modified asphalt, which is not conducive to fully understanding the anti-cracking behavior of modified asphalt at low temperature and the construction of durable pavement.

Currently, in order to obtain the microscale characterization mechanism of the modified asphalt material, it has become mainstream to use advanced micro-testing technology to characterize the performance evolution mechanism of asphalt materials [19]. The modification of polymers by synthetic carbon nanotubes was investigated. And the results showed that the distribution of carbon nanotubes in the elastomer structure appeared as a local aggregation phenomenon at the microscale [20]. In terms of testing technology, many studies have pointed out that atomic force microscopy (AFM) has significant advantages in nanoscale phase structure characterization of asphalt binders due to its high-resolution [21,22,23]. Moreover, for the composite-modified asphalt, the nanoscale phase structure and its aging behavior are also unclear.

Therefore, this study focuses on exploring tensile properties and nanoscale phase structures of different modified asphalts, and their aging behavior. For this, one virgin asphalt and three modified asphalts, namely, 4% SBS-modified asphalt, 2% SBS and 20% crumb rubber (CR) composite asphalt, and 4% SBS and 2%TiO_2_ composite asphalt, are prepared and investigated. The possible correlations between the results of tensile cohesion and phase structure characterization are discussed in order to provide a theoretical basis and data support for the performance evolution mechanism of modified asphalt materials.

## 2. Materials and Research Methods

### 2.1. Materials

#### 2.1.1. Properties of Raw Materials

One virgin asphalt with 70 penetration grade, one linear styrene–butadiene–styrene (SBS) modifier, and a 40-mesh crumb rubber (CR), obtained from an asphalt plant, were used in this study. In addition, one nano titanium dioxide (TiO_2_) was also used for obtaining the nanoparticle-modified asphalt. SBS copolymer has a block ratio of 30/70 and a weight-average molecular weight of 120,000 g/mol. The crumb rubber was processed from truck tires, which contain 55% natural rubber and synthesized rubber. TiO_2_ has an average particle size of 20 nm and a purity of 99.9%. TiO_2_ was treated with silicone oil on the surface to ensure the oil-wet properties of the material (specific disposal process information is not provided by the manufacturer) and the affinity of asphalt material. The appearance of various modifiers is shown in Figure 1.

#### 2.1.2. Modified Asphalt Preparation

Referring to the previous research on asphalt preparation methods [24], three kinds of modified asphalt (4% SBS-modified asphalt, 2% SBS and 20% crumb rubber (CR) composite asphalt, and 4% SBS and 2% TiO_2_ composite asphalt) were prepared based on the physical blending method. The content of each modifier and the conventional performance test results of modified asphalts are summarized in Table 1.

### 2.2. Aging Procedure

A thin-film oven and a custom-developed ultraviolet aging tester were used to simulate short-term thermo-oxygen aging (STA) and long-term thermo-oxygen–ultraviolet (UV) coupling aging (LTA), respectively. Considering that this study focuses on the aging behavior of modified asphalt, the STA test was conducted at 175 °C for 5 h. Ultraviolet aging testers have the ability to simulate thermo-oxygen–ultraviolet coupling aging conditions. It has a wavelength range of ultraviolet light from 300 to 400 nm. The aging temperature of the LTA test was set to 60 °C, and the aging time was set to 10 days. To avoid the aging gradient problem caused by a thicker asphalt film, the aged asphalt samples were poured at 20 g per plate. Notably, the LTA specimen must undergo the STA test.

### 2.3. Research Methods

#### 2.3.1. Force-Ductility Test

Force-ductility tests were conducted on the various asphalt specimens according to European Standard EN 13589 [9]. The specimens were prepared using straight molds (shown in Figure 2a). The straight mold is different from the “8”-shaped mold, which has a constant cross-section over the length being stretched. Before the testing, all specimens were kept at 5 °C in the SYD-4508G asphalt ductility instrument, Shanghai Changji Geological Instrument Co., Ltd., Shanghai, China. During the testing process, each specimen was stretched at a constant speed of 50 mm/min (shown in Figure 2b). Tensile force and ductility values were real-time recorded by the procedure. The test stopped when the specimen broke or the maximum elongation of 400 mm elongation was reached.

Three types of failure (brittle fracture, flow fracture, and ductile fracture) occur in force-ductility tests [13]. Figure 3 depicts the force-ductility results of virgin asphalt and SBS&CR-modified asphalt. It is obvious that flow fracture is seen for virgin asphalt, whereas ductile fracture is seen for SBS&CR asphalt. Concretely, the force-ductility curve of virgin asphalt consists of two stages: elastic stage (OA1) and yield stage (A1B1). However, the force-ductility curve of modified asphalt consists of three stages: elastic stage (OA2), stress yield stage (A2B2), and creep stage (B2C2). The creep stage is the special stage of modified asphalt, which is closely related to the type and content of the polymer. As shown in Figure 3, *F_max1_* and *F_max2_* represent the peak force of the virgin and modified asphalt binders, respectively. Generally, the peak load that can be determined by the virgin asphalt, and additional increases in strains, continuously decrease the load up to the point at which the polymeric phase is activated and starts resisting (point B2 in Figure 3). Hence, the decrease in load and the amount of strain required (denoted by the area under the curve from A2 to B2) for the activation of the polymeric phase denotes how promptly a polymer network responds to external loading. In this study, the area under the curve from A2 to B2 can be expressed as activation energy (*W*). After the polymer phase is activated, the specimen continues to be stretched until it breaks, at which point the ductility reaches its maximum (*L_max_*) and the corresponding tensile force is defined as fracture force (*F_f_*).

Moreover, for the polymer asphalt materials, tensile compliance and fracture toughness ratio were usually used to characterize the toughness properties in most studies. In this paper, tensile compliance (*f*) is defined as the ratio of maximum ductility value to peak force, denoted as *L_max_/F_p_*. And fracture toughness ratio (*R*) is the ratio of the area at creep stages to the areas at the elastic and stress yield stages in the force-ductility curve, expressed as (S _from A2 to D2_ − S _from A2 to Q_)/S _from A2 to Q_. In a word, many indicators can be obtained to evaluate the tensile properties of asphalt binders based on the force-ductility curve, such as peak force (*F_p_*), yield force (*F_y_*), fracture force (*F_f_*), fracture ductility (*L_max_*), tensile flexibility (*f*), activation energy (*W*), and toughness ratio (*R*). The definitions of these indicators are summarized in Table 2.

#### 2.3.2. Atomic Force Microscopy

A Dimension Icon AFM instrument made by Bruker was selected to obtain AFM mappings for evaluating the nanoscale phase structure of different kinds of asphalt. Considering the viscoelastic properties of the asphalt materials, an RTESPA-type probe with an elastic constant of 3 N/m was adopted. During the test, the test frequency was set to 1 Hz, and the scanning range was 20 × 20 μm. AFM test samples were prepared via the natural flow-forming method with metal trays according to the literature [25]. For each sample, 5 test areas were tested. With the aid of professional software attached to AFM, the root-mean-square roughness (*R_q_*) was calculated to characterize changes in the phase morphology.

The flow chart of this paper is shown in Figure 4.

## 3. Results and Discussion

### 3.1. The Evolution of Tensile Cohesion Properties

#### 3.1.1. Tensile Properties Characterization

As presented in Figure 5, all force-ductility curves start to show a linear increase in the tensile force value with increasing ductility until a peak force is obtained. However, there is a significant difference in the post-peak stretched behaviors between the virgin asphalt and modified asphalts. In the view of failure type, flow failure is only seen for virgin asphalt, while ductile fracture is found for the three modified asphalts. This is because when the tension force reaches its peak, the strong polar interaction between the molecules of the virgin asphalt components breaks down, thus reducing the resistance of the tensile flow. As ductility continues to increase, the force-ductility curve shows stress relaxation. However, when the polymer phase in asphalt is activated, the force-ductility curve exhibits creep because the polymer components can bear the tensile load. In other words, when the polymer phase is fully activated, the curves of modified asphalt start to enter the ductile flow stage until a fracture occurs. But the specific ductile flow behavior is determined by the polymer.

As can be seen from Figure 5, although the maximum ductility values of the three modified asphalts are similar, the ductile behavior of the asphalt binders shows a big difference in the tensile process. It can be clearly found that the tensile curve of SBS-modified asphalt is similar to that of SBS&TiO_2_-modified asphalt in shape. By comparison, the force-ductility curve of SBS&CR-modified asphalt has little variation in tensile force per unit elongation during the creep stage. This highlights that SBS&CR-modified asphalt has a better tensile toughness than other asphalts.

In order to better characterize the tensile cohesion behavior of different asphalts, some indicators are calculated and plotted, as shown in Figure 6 and Figure 7.

It can be found from Figure 6 that peak force *F_p_* and fracture force *F_f_* of SBS&CR-modified asphalt are smaller than that of SBS-modified asphalt and SBS&TiO_2_-modified asphalt. However, yield force *F_y_* of SBS&CR-modified asphalt is larger than that of the other two modified asphalts. *F_p_*, *F_y_*, and *F_f_* represent the tensile cohesive strength, yield strength, and the final failure strength corresponding to the maximum ductility, respectively. Hence, SBS&CR-modified asphalt has a smaller tensile cohesive strength in the elastic phase of the stretching process. But it has a larger yield strength, which highlights a better resistance to stress relaxation behavior. This also indirectly indicates that the polymer phase of SBS&CR-modified asphalt is the most easily activated. Combined with the results of activation energy *W* (shown in Figure 6b), it can be clearly seen that SBS&CR-modified asphalt has the smallest activation energy among the three modified asphalts. Generally speaking, *W* is believed to be an important indicator for understanding the transition phase (virgin asphalt to activate the polymeric matrix) of polymer-modified asphalt [17]. Hence, SBS&CR-modified asphalt is better than that of SBS-modified asphalt and SBS& TiO_2_-modified asphalt in terms of activation energy. By contrast, although the peak forces of SBS-modified asphalt and SBS&TiO_2_-modified asphalt are larger, those of the yield forces are significantly lower. Moreover, the activation energy of SBS-modified asphalt and SBS&TiO_2_-modified asphalt is 2.87 times and 3.31 times that of SBS&CR-modified asphalt, respectively. This indicates that the activation of the SBS polymer phase is more difficult or requires more energy to activate when the rubber powder is not present. So, there is no doubt that this result should be related to the interaction between rubber powder and asphalt. In addition, SBS&CR-modified asphalt has the lowest fracture force, which shows a better low-temperature performance.

Fracture ductility is the most common index to evaluate the tensile properties of asphalt. As shown in Figure 7a, although there is no significant difference in the fracture ductility values of the three modified asphalts, SBS&TiO_2_-modified asphalt has the maximum value. It shows that TiO_2_ modifiers can improve the ductility of SBS-modified asphalt at low temperature. In addition, the tensile flexibility (f) and toughness ratio (*R*) are widely accepted indexes to evaluate the low-temperature tensile toughness of asphalt materials in the force-ductility test. From Figure 7b,c, it can be seen that SBS&CR-modified asphalt has the largest tensile compliance and toughness ratio among the three modified asphalts. From this point, SBS&CR-modified asphalt has the best low-temperature tensile toughness. This may be attributed to rubbers released by the desulfurization and depolymerization of CR particles that make the binder more flexible at low temperatures. It is also reported that rubbers contribute more to the properties of crumb rubber/SBS-modified asphalt than SBS copolymers [18]. In addition, compared with the SBS-modified asphalt, SBS& TiO_2_-modified asphalt has a higher toughness ratio. This also proves that the addition of TiO_2_ modifiers is helpful for improving the toughness. However, taking into account the economic cost of raw materials, rubber powder additives have more application prospects than TiO_2_ for obtaining a better tensile toughness of SBS-modified asphalt at low temperature.

#### 3.1.2. Effect of Aging on the Tensile Cohesion Properties

Figure 8 presents the appearance changes in virgin asphalt and three modified asphalts under different aging conditions.

From Figure 8, it can be observed that the appearance of all the asphalts changed with the increase in aging level. Visible exterior damage can be clearly seen for LTA aged asphalt regardless of the type of asphalt. Thus, the aging resistance of different asphalt binders can be identified through the aging conditions set in this study. By comparison, it was found that the damage of virgin asphalt is the most serious, and many cracks (i.e., folding phenomenon) are found in the appearance image. Also, extensive damage is presented for SBS-modified asphalt and SBS&TiO_2_-modified asphalt. However, only minor local damage is found for SBS&CR-modified asphalt, which indicates a better aging resistance. In order to further explore and compare the effect of aging on the tensile cohesion properties of different asphalt binders, the force-ductility tests are conducted for all aged asphalts. The force-ductility curves of aged asphalt are plotted, as shown in Figure 9. And some tensile cohesion indicators are also calculated, as presented in Table 3.

For all asphalts, with the increase in aging level, the peak force tends to increase, while the fracture ductility tends to decrease. This shows that aging increases the tensile strength and decreases the ductility of asphalt. A closer look found that STA aging caused a significant increase in peak tension, while LTA aging had less of an effect on it regardless of the type of asphalt. By contrast, LTA aging has a more significant impact on the third stage (namely, the creep phase) in the tensile curves of the asphalts. This is because a higher thermal-oxygen aging temperature of the STA process makes the aging rate of the asphalt phase larger, resulting in an increase in the elasticity of the asphalt phase and an increase in tensile strength. While during the aging process of LTA, the polymer phase degradation is dominant, and the creep behavior of the third stage in the tensile curve is greatly affected.

The activation energy can be used to characterize the resistance response rate of the polymer phase in the stress relaxation stage of the tensile process, and also to reflect the interaction between the polymer phase and asphalt phase. As can be seen from Figure 9, regardless of the type of modified asphalt, STA aging increases the activation energy of the polymer phase, while LTA aging decreases the activation energy. Therefore, compared with unaged asphalt, aging delays the response rate of the polymer phase and reduces the interaction between the asphalt phase and the polymer. This is because the thermal-oxygen aging of the asphalt phase in the STA process is significant, which greatly reduces the interaction between the asphalt phase and the polymer phase, and increases the activation difficulty of the polymer phase to a certain extent. However, in the LTA process, part of the polymer phase is degraded, which reduces the relative content of the polymer phase in the asphalt, resulting in a decreasing trend of activation energy.

On the other hand, from the quantitative results of activation energy, the activation energy of the three modified asphalts is concentrated in 308~390 N·cm, 107~140 N·cm, and 354~582 N·cm, respectively. It is clear that the activation energy of SBS&CR-modified asphalt is much smaller than that of the other two asphalt under the same aging level. This indicates that the polymer phase in SBS&CR-modified asphalt is more easily activated during the tensile process and its tensile toughness is the best among the three modified asphalts under the same aging level. In addition, it is worth noting that the activation energy of SBS&TiO_2_-modified asphalt after STA is increased by 49.2% more than that of SBS-modified asphalt after STA. And the activation energy of the SBS&TiO_2_-modified asphalt after LTA is increased by 62% more than that of SBS-modified asphalt after LTA. This indicates that the addition of TiO_2_ increases the activation energy of the polymer phase, and it becomes more obvious with the increase in aging level. The reason for this phenomenon may be that with the increase in aging level, TiO_2_ particles have a more prominent shielding effect on the asphalt phase and SBS phase, which increases the activation difficulty of the polymer phase and reduces the response speed of the polymer phase during stretching.

Table 3 shows the tensile cohesion indicators of aged asphalt binders. For the three modified asphalts, fracture ductility, tensile compliance, and toughness ratio decrease gradually with the increase in aging level. This indicates that both STA and LTA reduce the tensile properties of asphalts and are prone to low-temperature cracking in pavements. Therefore, it is very necessary to compare the evolution behaviors of the tensile properties of different modified asphalts.

First of all, from the perspective of fracture ductility, SBS&TiO_2_-modified asphalt is larger than that of the other two asphalts under the same aging level, which shows that TiO_2_ can significantly improve the low-temperature ductility of SBS-modified asphalt. However, from the view of the percentage reduction in fracture ductility, SBS-modified asphalt is 6.1% and 15.7% after STA and LTA, while that of SBS&CR-modified asphalt is 5.6% and 0.3% and SBS&TiO_2_-modified asphalt is 2.9% and 5.4%, respectively. On this basis, TiO_2_ additives have a more prominent thermal-oxygen aging resistance, while rubber powder particles have better ultraviolet aging resistance.

On the other hand, from the view of tensile compliance and toughness ratio, SBS&CR asphalt and SBS&TiO_2_ asphalt are also significantly greater than those of SBS-modified asphalt under the same aging level. Taking the asphalt specimens after LTA as an example, the tensile compliance of SBS&CR asphalt and SBS&TiO_2_ asphalt is 1.64 times and 1.11 times that of SBS-modified asphalt, respectively. Similarly, the toughness ratio of SBS&CR asphalt and SBS&TiO_2_ asphalt is 1.25 times and 1.17 times that of SBS-modified asphalt, respectively. This again shows that the tensile properties of SBS&CR asphalt and SBS&TiO_2_ asphalt are better than those of SBS-modified asphalt under the same aging level. Hence, the addition of rubber powder particles and TiO_2_ in SBS-modified asphalt can improve the tensile toughness and aging resistance of SBS-modified asphalt. In fact, many studies have reported the excellent aging resistance of rubber powder-modified asphalt and nanoparticle-modified asphalt. It has been reported that the characteristics of inorganic nanoparticles such as high activity and high surface energy make it possible to improve the aging resistance of asphalt [26]. However, according to the results of tensile properties in this paper, and taking into account aging resistance and cost-effectiveness, SBS&CR asphalt is recommended as a pavement material in areas with strong ultraviolet radiation to prevent pavement cracking caused by aging asphalt materials at low temperature.

### 3.2. Effect of Aging on Nanoscale Phase Structure

#### 3.2.1. Nanoscale Phase Structure

Figure 10 shows the AFM mappings of different asphalts. As can be seen from Figure 10, bee phase structure can be found in all asphalt specimens. However, the size and number of the bee phase in each asphalt vary greatly. It is clearly found that the characteristics and distribution of bee phases in virgin asphalt and SBS asphalt are similar, while the bee phase number increases, and bee phase distribution is more uniform in SBS&TiO_2_ asphalt. However, it is more unexpected that the bee phase size in SBS&CR asphalt is the smallest, and its phase distribution is significantly different from other asphalts. Also, a closer inspection reveals that there are some incomplete bee phases, and some white and black spots like the components of bee phases are found in the SBS&CR asphalt. This phenomenon indicates that the SBS&CR-modified asphalt reaches a new phase equilibrium state due to the interaction between components. It can be inferred that the addition of rubber powder has a strong effect on the bee phase in asphalt. This may be due to the swelling and developmental behavior of rubber powder particles, which interfere with the formation of bee phase morphology. For example, the absorption of oil in asphalt by rubber powder particles destroys the original colloidal structure type of virgin asphalt.

By contrast, SBS modifier has little effect on the phase structure of virgin asphalt because a larger similarity is found between the virgin asphalt and SBS-modified asphalt. Moreover, TiO_2_ modifier increases the number of bee phases, and their distribution is more uniform, indicating the formation of a more stable phase structure system. This is in agreement with previous research [26], in which the larger specific surface area of TiO_2_ promotes the modified asphalt to form a more stable modified system. This may have contributed to its better tensile properties and aging resistance. From the above analysis, the phase structure of modified asphalt is related to the interaction between modifier components and virgin asphalt. Therefore, it is necessary to discuss the evolution behavior of the phase structure of modified asphalt binders under the influence of aging. It may be helpful to explain the evolution behavior of asphalt tensile properties.

#### 3.2.2. Effect of Aging on the Nanoscale Phase Structure

It is believed that roughness is a reasonable index to characterize the nanoscale phase structure of asphalt. So, the root mean square roughness (*Rq*) is calculated with the help of Nano-scope analysis software1.41 included with AFM. AFM mappings and roughness results of aged asphalts are shown in Figure 11.

Figure 11a shows that except for SBS&CR asphalt, virgin, SBS asphalt, and SBS&TiO_2_ asphalt, all have a bee phase structure in AFM mappings during the whole aging process. However, with the increase in aging level, the size of the bee phase has a tendency to decrease, especially for the asphalt specimens after LTA. Also, for the surface roughness *Rq*, the three asphalts after STA reduced by 18.4%, 4.7%, and 12.9%, and that of the three asphalts after LTA reduced by 18.2%, 6.3%, and 42.6%, respectively. So, aging reduces the height difference between the phases in the asphalt binders and makes the surface in the nanoscale flatter. By comparison, the reduction ratio of surface roughness of SBS-modified asphalt is significantly smaller than that of virgin asphalt. This is because the degradation of SBS and the asphalt phase aging co-occur during the whole SBS-modified asphalt aging process. Although both of them will lead to a reduction in surface roughness, the aging rate of the asphalt phase is greatly reduced due to the presence of SBS. For the SBS&TiO_2_ asphalt, apart from asphalt phase aging and SBS degradation, TiO_2_ molecules inhibit the aggregation behavior of polar molecules during the aging process, which also leads to the reduction in surface roughness. After LTA, the interaction force between polar molecules in asphalt increased, and there was a trend of aggregation. At this time, when TiO_2_ is present, a large amount of asphaltene will be adsorbated around the nano TiO_2_ clusters, thus preventing the formation of large-sized polar molecular aggregation structures. This may be the main reason for the significant reduction in surface roughness for SBS&TiO_2_ asphalt.

Different from other asphalts, the phase structure evolution behavior of SBS&CR-modified asphalt is more complex. As can be seen from Figure 11, some bee phase structures in AFM mapping show a trend of growing after STA, while phase structures similar to black and white spots gradually increase. Also, the surface roughness increases from 1.2 nm to 2.3 nm (increased by 91.6%). Further, after LTA, the bee phase size decreases significantly, but the bright white spot phase continues to increase, and the surface roughness decreases from 2.3 nm to 1.1 nm (close to the unaged asphalt). This shows that the interaction between rubber powder and asphalt components in SBS&CR asphalt continues during the aging process, resulting in a large mutation in phase structure. STA aging induces asphalt phase aging, resulting in part of the bee phase size increase, which is the main reason for the increase in surface roughness. At the same time, the higher temperature of STA aging also promotes the swelling effect of rubber powder, which continues to release polymers, carbon black, antioxidants, and other substances into the asphalt phase, making the AFM mappings show a black and white spot new phase structure. After LTA, carbon black, antioxidants, and other substances gradually decompose to form a shielding effect on the asphalt phase, and once again improve the aging resistance of the asphalt phase. The desulfurization and degradation behavior of rubber powder also continues, which reduces the cross-linking density of rubber powder, promotes compatibility between phase states, and further reduces surface roughness. According to the literature [27], the swelling, desulfurization, and degradation of rubber powder particles and asphalt components in the process of blending and aging results in the disappearance of the bee phase structure. Therefore, the interaction of the rubber powder phase and the asphalt phase is the leading cause of the evolution of the modified asphalt phase structure.

It has been reported that surface roughness represents the surface height difference among the nanoscale phases [28]. The lower the surface roughness value, the smaller the surface height difference among the nanoscale phases, and the flatter the asphalt nanoscale surface. Thus, there is no doubt that the surface of SBS&CR-modified asphalt is flatter than that of other asphalts because it has the lowest surface roughness under the same aging condition. The smaller the difference between the phases, the less likely it is to produce phase stress concentration during the tensile process. This may be the micro reason for its good macro-scale tensile toughness. Actually, many studies have reported that CR-modified asphalt has a better resistance to aging [24,29,30]. However, it is believed that higher roughness characterizes more abundant nanostructures, indicating a symbol of excellent performance [26]. Based on the results of this study, this conclusion may only be applicable to the comparison of the same asphalt, and the roughness comparison between different types of modified asphalt may not be of great significance.

## 4. Conclusions

In order to explore tensile properties, nanoscale phase structures of different modified asphalts, and their aging behavior, one virgin asphalt and three modified asphalts, namely, 4% SBS-modified asphalt, 2% SBS & 20% crumb rubber (CR) composite asphalt, and 4% SBS & 2%TiO_2_ composite asphalt, were prepared and investigated using the force-ductility test and atomic force microscopy (AFM). The main conclusions are as follows:(1)Ductile fracture is found for the three modified asphalts. However, for the activation energy, SBS asphalt and SBS&TiO_2_ asphalt are 2.87 times and 3.31 times than that of SBS&CR asphalt, respectively. This demonstrates that the activation of the SBS polymer phase requires more energy during the stretching process when the rubber powder is not present.(2)SBS&TiO_2_ asphalt has the maximum fracture ductility value, which shows that TiO_2_ modifiers can improve the ductility of SBS-modified asphalt at low temperature. However, SBS&CR-modified asphalt has the largest tensile compliance and toughness ratio. The reason for this phenomenon is that rubbers released by the desulfurization and depolymerization of CR particles make the binder more flexible at low temperature.(3)SBS&CR asphalt and SBS&TiO_2_ asphalt show better tensile properties and aging resistance in terms of the quantitative results of tensile property indicators, indicated by a larger value of fracture ductility, tensile compliance, and the toughness ratio under the same aging condition.(4)According to the AFM results, SBS modifier has little effect on the phase structure of virgin asphalt, while TiO_2_ modifier increases the number of bee phases and makes their distribution more uniform, indicating the formation of a more stable phase structure system. This may contribute to its better tensile properties and aging resistance. Moreover, TiO_2_ molecules inhibit the aggregation behavior of polar molecules during the aging process, which leads to a reduction in surface roughness.(5)SBS modifier has little effect on the phase structure of virgin asphalt, while TiO_2_ modifier increases the number of bee phases and makes their distribution more uniform, indicating the formation of a more stable phase structure system. Also, TiO_2_ molecules inhibit the aggregation behavior of polar molecules during the aging process, which leads to a reduction in surface roughness. This may have contributed to its better tensile properties and aging resistance. By comparison, the effect of aging on the phase structure of SBS&CR asphalt is more significant among the three modified asphalts. This result can be attributed to the interaction between rubber powder particles and asphalt.

## Figures and Tables

**Figure 1 polymers-16-03121-f001:**
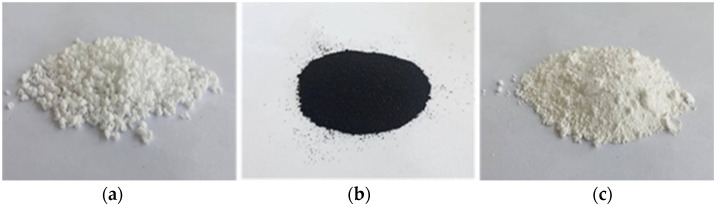
Appearance of various modifiers: (**a**) SBS; (**b**) CR; (**c**) TiO_2_.

**Figure 2 polymers-16-03121-f002:**
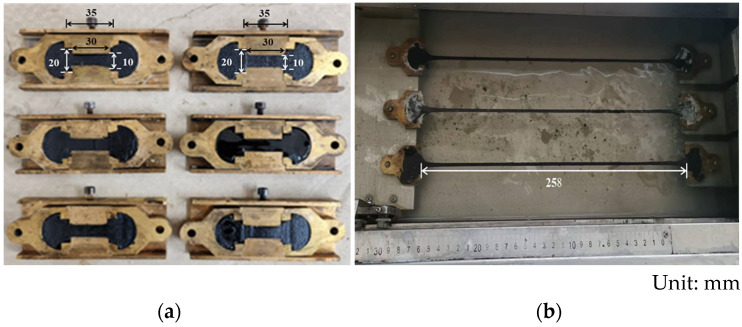
Specimens and tensile process in the force-ductility test: (**a**) Specimens; (**b**) Tensile process.

**Figure 3 polymers-16-03121-f003:**
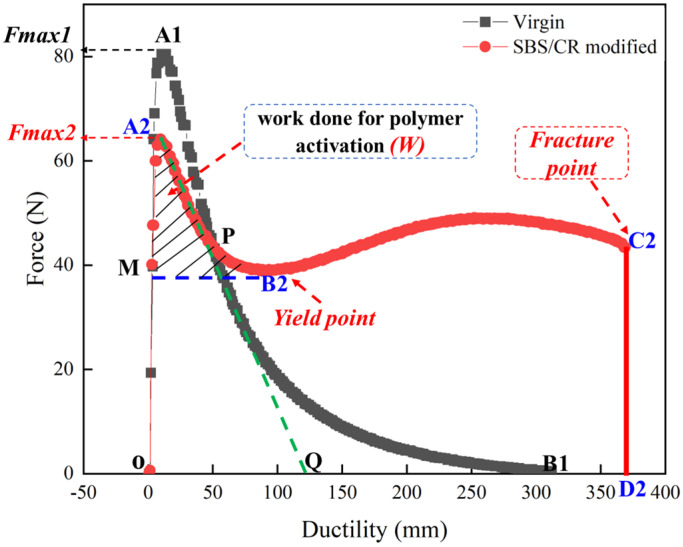
Example of force−ductility curves of asphalt binders.

**Figure 4 polymers-16-03121-f004:**
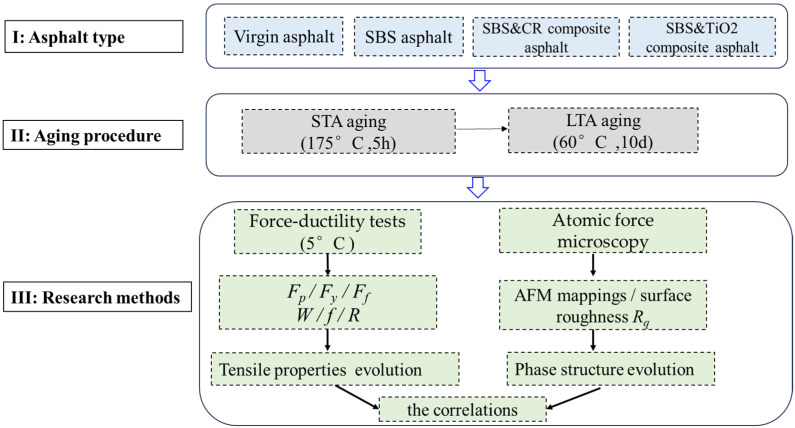
Flow chart of this paper.

**Figure 5 polymers-16-03121-f005:**
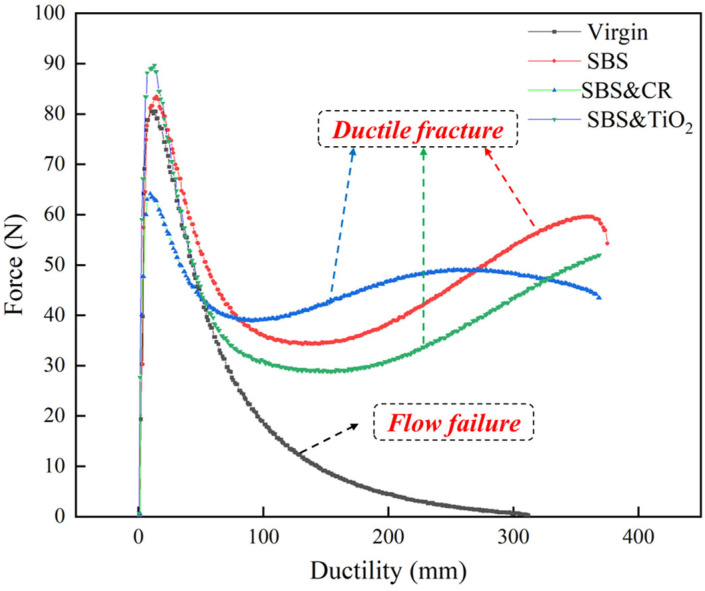
Force-ductility curves of different asphalt binders.

**Figure 6 polymers-16-03121-f006:**
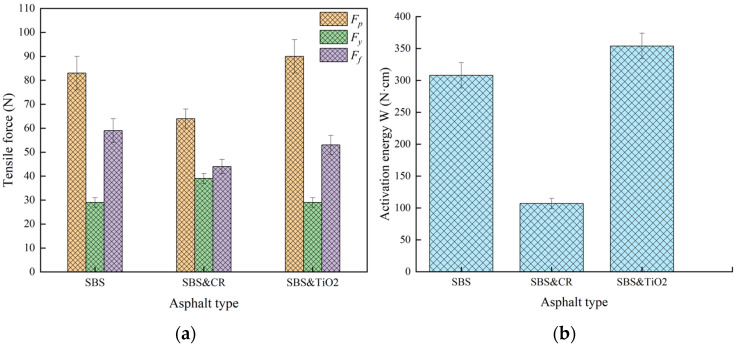
Results of tensile strength and polymer phase activation energy: (**a**) Peak force (*F_p_*), yield force (*F_y_*), and fracture force (*F_f_*); (**b**) Activation energy (*W*).

**Figure 7 polymers-16-03121-f007:**
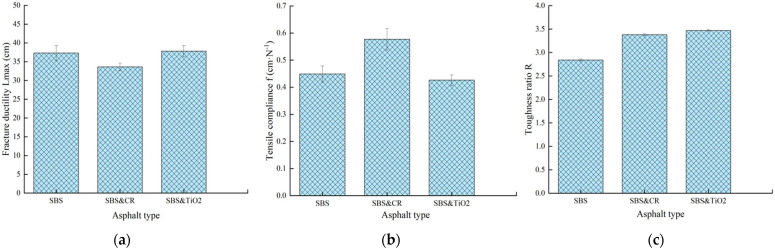
Tensile cohesion characterization of different asphalts: (**a**) Fracture ductility; (**b**) Tensile flexibility; (**c**) Toughness ratio.

**Figure 8 polymers-16-03121-f008:**
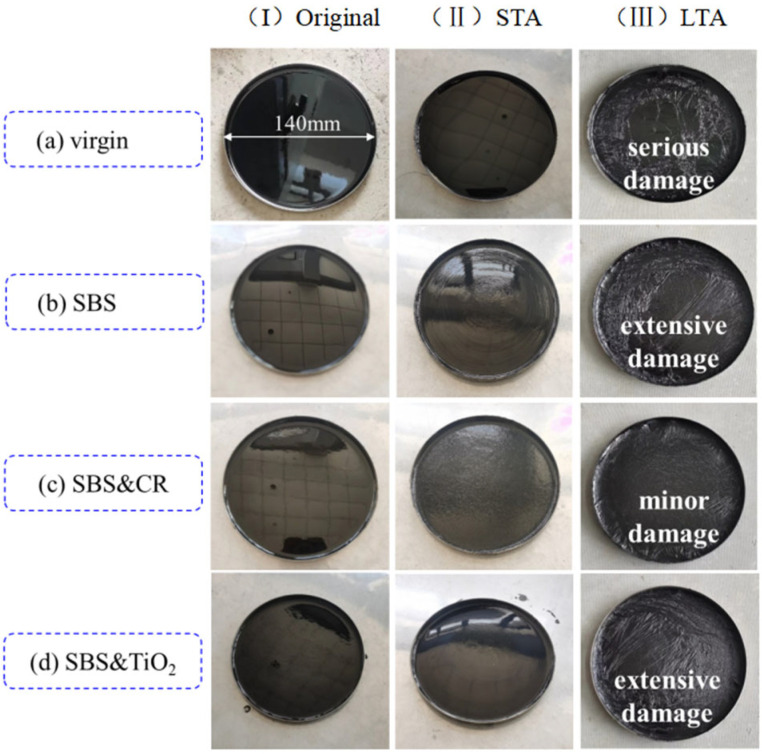
Appearance changes in aged asphalt.

**Figure 9 polymers-16-03121-f009:**
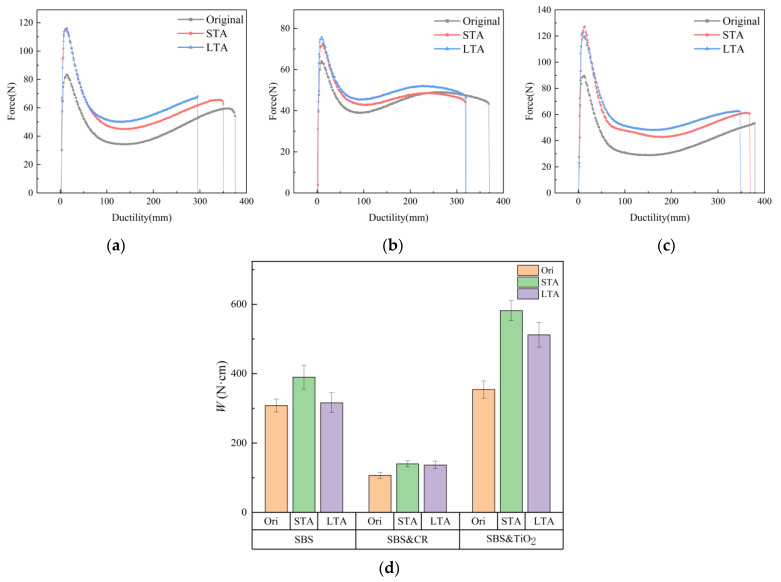
Force-ductility curves and activation energy of various modified asphalts: (**a**) SBS asphalt; (**b**) SBS&CR asphalt; (**c**) SBS&TiO_2_ asphalt; (**d**) activation energy.

**Figure 10 polymers-16-03121-f010:**
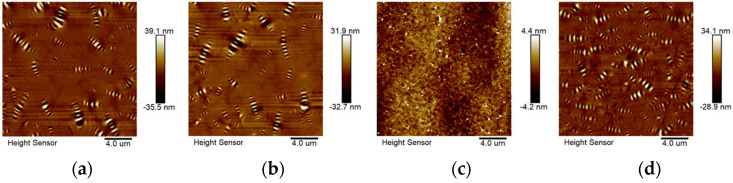
AFM mappings of different asphalts: (**a**) Virgin asphalt; (**b**) SBS asphalt; (**c**) SBS&CR asphalt; (**d**) SBS&TiO_2_ asphalt.

**Figure 11 polymers-16-03121-f011:**
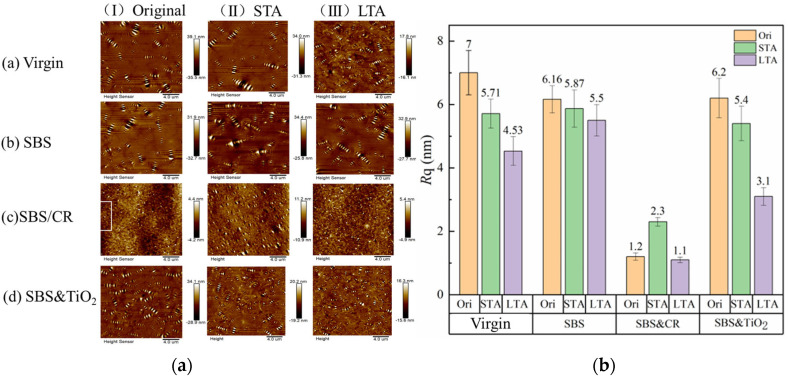
AFM mappings and root mean square roughness of aged asphalts. (**a**) AFM mappings; (**b**) Root mean square roughness *Rq*.

**Table 1 polymers-16-03121-t001:** The conventional properties of the modified asphalts.

Modifier Type and Content			Asphalt Type/Identification Code	Conventional Properties			
SBS (%)	CR (%)	TiO_2_ (%)		Penetration at 25 °C (mm^−1^)	Softening Point (°C)	Ductility (5 °C, cm)	Elastic Recovery (%)
0	0	0	Virgin asphalt/Virgin	65	49	31.19	-
4			SBS asphalt/SBS	57.3	81.3	37.28	84
2	20		SBS&CR composite asphalt/SBS&CR	56.2	76.1	33.64	83
4		2	SBS&TiO_2_ composite asphalt/SBS& TiO_2_	58.1	82.1	37.81	81

**Table 2 polymers-16-03121-t002:** The definition of evaluation indicators based on force-ductility test.

Indicators	Unit	Definition
*F_p_*	N	The force corresponding to the peak point (A1&A2) in Figure 3
*F* * _y_ *	N	The force corresponding to the yield point (B1&B2) in Figure 3
*F_f_*	N	The force corresponding to the fracture point (D2) in Figure 3
*L_max_*	cm	Defined as maximum ductility value
*f*	cm·N^−1^	Defined as the ratio of maximum ductility value to peak force, denoted as *L_max_*/*F_p_*
*W*	N·cm	Defined as the area under the curve from A2 to B2, denoted as *W* (the area of the slash-shaded area in Figure 3)
*R*	/	Defined as the ratio of the area under the curve, denoted as (S _from A2 to C2_ − S _OA2Q_)/S _OA2Q_

**Table 3 polymers-16-03121-t003:** Tensile cohesion indicators of aged asphalt binders.

Asphalt Type	Aging Level	*L_max_*/mm	*f*/cm·N^−1^	*R*
SBS	Ori	373	0.449	2.84
	STA	350	0.301	2.77
	LTA	295	0.257	2.35
SBS&CR	Ori	336	0.577	3.38
	STA	317	0.451	2.96
	LTA	316	0.421	2.94
SBS&TiO_2_	Ori	378	0.426	3.47
	STA	367	0.315	3.08
	LTA	347	0.284	2.76

## Data Availability

Some or all data, models, or codes that support the findings of this study are available from the corresponding author upon reasonable request.

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
