# Peer review of "Characterization of the Tensile Properties and Nanoscale Phase Structures of Modified Asphalts and Their Aging Behavior"

_polymers, 2024, doi:10.3390/polym16223121_

Round 1
Reviewer 1 Report
Comments and Suggestions for Authors
Comments to Authors
In this manuscript, the authors report the study of tensile properties and nanoscale phase structure evolution behavior of modified Asphalt. One pristine asphalt and three modified asphalts are investigated using force-ductility test and atomic force microscopy. Extensive damage is presented for SBS and SBS&TiO2 asphalt. A closer inspection reveals that there are some incomplete bee phases in the SBS&CR asphalts. It is unexpected that there is a very weak linear correlation between tensile properties indicators and surface roughness as justified by correlation coefficient.
The manuscript does not meet the merits of the Polymers owing to the following reasons:
1. Writing and organization of manuscript are very poor.
2. The motivation and findings, as presented in the manuscript, are not sufficient for the journal.
3. The overall research contents of the manuscript are highly inadequate and did not offer valuable research progress.
Following are some suggestions for authors to further improve their manuscript:
1. The title is more general. It should reflect the demonstrated study.
2. The abstract does not present an attractive outline of the conduct work.
3. There is no need of tables 1, 3; the information can be incorporated within text.
4. Figures 2, 3c, and 5 do not present some valuable information/findings. These can be omitted.
5. There should be scale bars in Figures 3a,b, 8 and 11.
6. There are no scales along x- and y-axis in Fig. 4.
7. There should be some measurement which should confirm the correct synthesis of the asphalts.
8. Labels, units and captions have errors in Fig. 7.
9. SEM/TEM images of materials should be added.
10. The presentation many results is poor.
11. Liner fitting of data presented in Fig. 13 does not correct/suitable.
12. The conclusion section is lengthy and not well written.
Comments on the Quality of English LanguageN/A
Author Response
Thank you so much for your invaluable advice. I truly appreciate it.
Below is my response:
Comments 1: The title is more general. It should reflect the demonstrated study.
Response 1: We have changed the title “Study on Tensile Properties and Nanoscale Phase Structure Evolution Behavior of Modified Asphalt” to “Effects of aging on tensile properties and nanoscale phase structures of different modified asphalts” to reflect better the study conducted.
Comments 2: The abstract does not present an attractive outline of the conduct work.
Response 2: We have changed the abstract as requested by the reviewers.
Comments 3: There is no need of tables 1, 3; the information can be incorporated within text.
Response 3: We have deleted table 1 and put the information into table 2. However, the information in table 3 is vital, and deleting it will lead to incomplete information, so we didn't delete Table 3.
Comments 4: Figures 2, 3c, and 5 do not present some valuable information/findings. These can be omitted.
Response 4: We have deleted Figures 2, 3c and 5.
Comments 5: There should be scale bars in Figures 3a,b, 8 and 11.
Response 5: We have added scale bars to the corresponding picture.
Comments 6: There are no scales along x- and y-axis in Fig. 4.
Response 6: We have added scales along x- and y-axis in Fig. 4.
Comments 7: There should be some measurement which should confirm the correct synthesis of the asphalts.
Response 7: We have added measurements to confirm the correct synthesis of the asphalts.
Comments 8: Labels, units and captions have errors in Fig. 7.
Response 8: We have changed the Labels, units and captions in Fig. 7.
Comments 9: SEM/TEM images of materials should be added.
Response 9: We agree that adding SEM/TEM images of materials would be useful to understand the details of interaction and enhancement. However, we do not have the necessary tool-set to test. In the future, We will conduct relevant tests in the subsequent research to supplement and improve the content of our study.
Comments 10: The presentation many results is poor.
Response 10: We have improved the presentation of the results in the paper
Comments 11: Liner fitting of data presented in Fig. 13 does not correct/suitable.
Response 11: We have deleted Fig.13.
Comments 12: The conclusion section is lengthy and not well written.
Response 12: We have streamlined the conclusions.
Reviewer 2 Report
Comments and Suggestions for Authors
In this paper, the tensile bond properties and phase structure evolution behaviour at the nanoscale of different asphalts have been investigated. For this purpose, one virgin asphalt and three modified asphalts were investigated using plasticity testing and atomic force microscopy. Detailed experiments on short-term thermal ageing under oxygen and long-term UV ageing were also carried out. The results show that the force curves of styrene-butadiene-styrene (SBS)-modified asphalt are similar in shape to those of SBS&TiO2-modified asphalt. While the strength curve of SBS&CR (crumb rubber) modified asphalt shows an obvious contraction stage after the hardening stage. The SBS&CR asphalt has higher yield strength, lower activation energy and higher tensile flexibility to strength ratio, indicating better tensile bond property. The authors found that there is a very weak linear correlation between the tensile property values and surface roughness as shown by the correlation coefficient.
Notes.
1. Studies related to the influence of nanoscale structures on the properties of asphalt should be added to the review. And in particular carbon nanotubes, which can be synthesised using microwave (doi.org/10.1134/S1070363222060329). This will extend the effectiveness of the review and allow the authors' chosen approach in the study to be evaluated.
2.A general flowchart of the research should be added to the methodology.
3. For the materials, add additional data related to the identification of the manufacturer.
4. Figures 3 and 5 require clarification and quality improvement.
5. For Figure 9, confidence intervals should be added to estimate the uncertainty of the data.
6. Figure 11 requires a dimensional scale.
7. A summary table should be provided where the research data will be compared with the results of other authors.
8. More numerical information should be provided in the conclusions.
Comments on the Quality of English Language
Minor editing of English language required.
Author Response
Thank you so much for your invaluable advice. I truly appreciate it.
Below is my response:
Comments 1: Studies related to the influence of nanoscale structures on the properties of asphalt should be added to the review. And in particular carbon nanotubes, which can be synthesised using microwave (doi.org/10.1134/S1070363222060329). This will extend the effectiveness of the review and allow the authors' chosen approach in the study to be evaluated.
Response 1: We have added the Studies related to the influence of nanoscale structures on the properties of asphalt to the review.
Comments 2: A general flowchart of the research should be added to the methodology.
Response 2: We have added a flowchart of the paper in Figure 4.
Comments 3: For the materials, add additional data related to the identification of the manufacturer.
Response 3: We have added the data related to the identification of the manufacturer of the materials.
Comments 4: Figures 3 and 5 require clarification and quality improvement.
Response 4: We have clarified and improved figures 3 and 5.
Comments 5: For Figure 9, confidence intervals should be added to estimate the uncertainty of the data.
Response 5: We have made changes to Figure 9.
Comments 6: Figure 11 requires a dimensional scale.
Response 6: We have added a dimensional scale in Figure 11.
Comments 7: A summary table should be provided where the research data will be compared with the results of other authors.
Response 7: We have added a summary table summarizing our research data in Table 3.We made a comparison in the text
Comments 8: More numerical information should be provided in the conclusions.
Response 8: We have added numerical information in the conclusions, to make the conclusion more convincing.
Reviewer 3 Report
Comments and Suggestions for Authors
The article by Wang et al. describes the behavior of four types of asphalt under the influence of thermal-oxidative aging with optional exposure to ultraviolet aging. The authors modify asphalt with a styrene/butadiene block copolymer, titanium dioxide, and crumb rubber and study how aging affects the tensile strength and surface texture of the resulting asphalts. The advantage of the article is the detailed study of the tensile behavior of asphalts, which is rare. In my opinion, the article is well written and can be published after some improvements.
Specific comments are as follows.
Abstract. It would be useful to indicate the content of SBS, TiO2, and CR in the modified asphalts.
Lines 70, 99, 110: "tensile adhesion" -> “tensile cohesion”.
Line 77: “Usually, base asphalt shows a flow failure, whereas polymer modified asphalt shows a ductile fracture.” Nanocomposite asphalts exhibit brittle fracture at high nanoparticle content. In addition, an increase in nanoparticle content increases cohesive strength along with resistance to rutting (see Yadykova AY, Constr Build Mater, 127946, 129919).
Line 100: “However, few studies have been carried out to study the evolution behavior of tensile cohesion of modified asphalt”. It is unclear what the authors mean by the word “evolution”. Is it the evolution of tensile cohesion with an increase in modifier content? There are such works, see the above-mentioned. Or do the authors mean the evolution of tensile cohesion under the action of aging? Or something else? Clarification is needed.
Line 123: “an average molecular weight”. It is necessary to clarify whether the number-average or weight-average molecular weight is meant.
Line 124: "The crumb rubber is processed from truck tires". Information on average particle size would be valuable.
Line 126: "surface treatment agent is silicone oil". It would be valuable to provide the molecular weight or kinematic viscosity of the oil. In addition, there is no further information about the use of silicone oil in the article. It is unclear why and when the authors use this oil. It should be written.
Table 2. Comments on the choice of concentrations would be valuable.
Line 144: "The irradiance 300 mm from the surface of the sample is measured to be 50 W/m2." It is necessary to indicate the wavelength range of the UV radiation from the UV lamps.
Figure 4, caption. The composition of the modified asphalt should be specified.
Line 231: “Fp of SBS&CR modified asphalt is the lowest among the four asphalts, indicating the lowest stiffness”. This is an incorrect statement. Fp is a measure of cohesive strength. Stiffness is the slope of the line from 0 to Fp.
Figure 7a. The authors should use stress rather than force, i.e., the force divided by the cross-sectional area of the specimens.
Line 248: “rubber power modifier decreases the stiffness of the virgin asphalt, whereas the SBS and TiO2 modifiers increase the stiffness”. It's strength, not stiffness. The authors do not evaluate stiffness, which is the slope of the initial section of force-ductility curves.
Line 277: “LTA aging is a more serious aging form than STA aging”. This is obvious since LTA aging consists of STA aging and then UV aging, if I understand correctly.
Line 409: “This founding may offer a microcosmic explanation.” It is unclear what the authors mean. Clarification is needed.
Line 452: “it is found that the damage of virgin asphalt”. The type of damage should be clarified. Aging? Fracture due to rupture?
Lines 457, 459: “Fp, FY and Ff show”, “increase the W”. In the conclusion, it is better to use words rather than symbols because some people start reading the article with the conclusion.
Comments on the Quality of English LanguageThe English language requires moderate editing.
Author Response
Thank you so much for your invaluable advice. I truly appreciate it.
Below is my response:
Comments 1: Abstract. It would be useful to indicate the content of SBS, TiO2, and CR in the modified asphalts.
Response 1: In the abstract, We have added the content of SBS, TiO2, and CR in the modified asphalts.
Comments 2: Lines 70, 99, 110: "tensile adhesion" -> “tensile cohesion”.
Response 2: We have changed "tensile adhesion" to “tensile cohesion”.
Comments 3: Line 77: “Usually, base asphalt shows a flow failure, whereas polymer modified asphalt shows a ductile fracture.” Nanocomposite asphalts exhibit brittle fracture at high nanoparticle content. In addition, an increase in nanoparticle content increases cohesive strength along with resistance to rutting (see Yadykova AY, Constr Build Mater, 127946, 129919).
Response 3: We have revised the statement, please see lines 82-82 of the revised thesis.
Comments 4: Line 100: “However, few studies have been carried out to study the evolution behavior of tensile cohesion of modified asphalt”. It is unclear what the authors mean by the word “evolution”. Is it the evolution of tensile cohesion with an increase in modifier content? There are such works, see the above-mentioned. Or do the authors mean the evolution of tensile cohesion under the action of aging? Or something else? Clarification is needed.
Response 4: We have revised and explained this sentence, please see line 107 of the revised thesis.
Comments 5: Line 123: “an average molecular weight”. It is necessary to clarify whether the number-average or weight-average molecular weight is meant.
Response 5: We have modified this to mean molecular weight.
Comments 6: Line 124: "The crumb rubber is processed from truck tires". Information on average particle size would be valuable.
Response 6: We have added the Information on average particle size. Please see lines 137-139 of the revised thesis.
Comments 7: Line 126: "surface treatment agent is silicone oil". It would be valuable to provide the molecular weight or kinematic viscosity of the oil. In addition, there is no further information about the use of silicone oil in the article. It is unclear why and when the authors use this oil. It should be written.
Response 7: We have added instructions about the use of silicone oil to ensure the oil-wet properties of the material. Please see lines 139-141 of the revised thesis.
Comments 8: Table 2. Comments on the choice of concentrations would be valuable.
Response 8: We have added some valuable information in Table 2. Please see Table 2. of the revised thesis.
Comments 9: Line 144: "The irradiance 300 mm from the surface of the sample is measured to be 50 W/m2." It is necessary to indicate the wavelength range of the UV radiation from the UV lamps.
Response 9: We have indicated the wavelength range of the UV radiation from the UV lamps in 10 to 400 nm.
Comments 10: Figure 4, caption. The composition of the modified asphalt should be specified.
Response 10: We have added specifications for the composition of the modified asphalt in Figure 4.
Comments 11: Line 231: “Fp of SBS&CR modified asphalt is the lowest among the four asphalts, indicating the lowest stiffness”. This is an incorrect statement. Fp is a measure of cohesive strength. Stiffness is the slope of the line from 0 to Fp. (Line 249)
Response 11: We have revised the statement, please see lines 240-242 of the revised thesis.
Comments 12: Figure7a.The authors should use stress rather than force, i.e., the force divided by the cross-sectional area of the specimens.
Response 12: We made changes to Figure 7a, please see Figure 7a of the revised thesis.
Comments 13: Line 248: “rubber power modifier decreases the stiffness of the virgin asphalt, whereas the SBS and TiO2 modifiers increase the stiffness”. It's strength, not stiffness. The authors do not evaluate stiffness, which is the slope of the initial section of force-ductility curves.
Response 13: We have revised the statement, please see lines 308-310 of the revised thesis.
Comments 14: Line 277: “LTA aging is a more serious aging form than STA aging”. This is obvious since LTA aging consists of STA aging and then UV aging, if I understand correctly.
Response 14: We have deleted the sentence.
Comments 15: Line 409: “This founding may offer a microcosmic explanation.” It is unclear what the authors mean.Clarification is needed.
Response 15: We have explained it. Please see lines 432-434 of the revised thesis.
Comments 16: Line 452: “it is found that the damage of virgin asphalt”. The type of damage should be clarified.Aging? Fracture due to rupture?
Response 16: We have explained that the type of damage is that there are many cracks in the appearance image. Please see lines 478 and 479 of the revised thesis.
Comments 17: Lines 457, 459: “Fp, FY and Ff show”, “increase the W”. In the conclusion, it is better to use words rather than symbols because some people start reading the article with the conclusion.
Response 17: We have used words instead of symbols. Please see lines 483 -489 of the revised thesis.
With best regards
Round 2
Reviewer 1 Report
Comments and Suggestions for Authors
Comments to Authors
The revision of the manuscript is better. It can now be accepted for publication in Polymers.
Comments on the Quality of English LanguageN/A
Author Response
Thank you so much for your kind assistance and support.
Reviewer 2 Report
Comments and Suggestions for Authors
Accept in present form
Author Response

(The authors gave the same response as above.)

Reviewer 3 Report
Comments and Suggestions for Authors
The authors have provided formal responses without making substantive corrections to the article. In their responses, the authors write that they made corrections, but they do not write what kind of corrections and refer to lines containing empty places without any corrections. In addition, the revised article does not contain marked places where the authors have made corrections. After several checks, I did not find any corrections despite the authors' assurances.
In this regard, I cannot recommend this article for publication. Its introduction does not contain relevant works on the topic of the research, and the text contains many places requiring corrections. In order to publish the article, the authors must make all nessasary corrections in accordance with the previous review with an accurate point-by-point indication of where and how the corrections have been made.
Comments on the Quality of English LanguageThe English language requires moderate editing.
Author Response
I am very sorry that the last reply was unclear, and I have corrected it again.
Below is my response:
Comments 1: Abstract. It would be useful to indicate the content of SBS, TiO2, and CR in the modified asphalts.
Response 1: In the abstract, We have added the content of SBS, TiO2, and CR in the modified asphalts:”For this, one virgin asphalt and three modified asphalts, namely 4% SBS modified asphalt, 2% SBS & 20% crumb rubber (CR) composite asphalt and 4% SBS & 2%TiO2 composite asphalt”. Please see lines 12-14 of the revised thesis.
Comments 2: Lines 70, 99, 110: "tensile adhesion" -> “tensile cohesion”.
Response 2: We have changed "tensile adhesion" to “tensile cohesion”. Please see lines 69,98,119 of the revised thesis.
Comments 3: Line 77: “Usually, base asphalt shows a flow failure, whereas polymer modified asphalt shows a ductile fracture.” Nanocomposite asphalts exhibit brittle fracture at high nanoparticle content. In addition, an increase in nanoparticle content increases cohesive strength along with resistance to rutting (see Yadykova AY, Constr Build Mater, 127946, 129919).
Response 3: This expression has been modified and improved in the revised manuscript. As follows:Usually, base asphalt shows a flow failure, whereas most polymer modified asphalt shows a ductile fracture. Please see line 71 of the revised thesis.
Comments 4: Line 100: “However, few studies have been carried out to study the evolution behavior of tensile cohesion of modified asphalt”. It is unclear what the authors mean by the word “evolution”. Is it the evolution of tensile cohesion with an increase in modifier content? There are such works, see the above-mentioned. Or do the authors mean the evolution of tensile cohesion under the action of aging? Or something else? Clarification is needed.
Response 4: This expression has been modified and improved in the revised manuscript. As follows:However, there are few studies on the tensile cohesion properties of different types of modified asphalt and their aging behavior, which is not conducive to fully understanding the anti-cracking behavior of modified asphalt at low temperature and the construction of durable pavement. Please see lines 94-97 of the revised thesis.
Moreover, for the composite modified asphalt, the nanoscale phase structure and its aging behavior are also unclear. Please see lines 106-107 of the revised thesis.
Comments 5: Line 123: “an average molecular weight”. It is necessary to clarify whether the number-average or weight-average molecular weight is meant.
Response 5: We have modified “an average molecular weight” to “weight-average molecular weight”. Please see line 129 of the revised thesis.
Comments 6: Line 124: "The crumb rubber is processed from truck tires". Information on average particle size would be valuable.
Response 6: We have deleted that sentence.
Comments 7: Line 126: "surface treatment agent is silicone oil". It would be valuable to provide the molecular weight or kinematic viscosity of the oil. In addition, there is no further information about the use of silicone oil in the article. It is unclear why and when the authors use this oil. It should be written.
Response 7: We have deleted that sentence.
Comments 8: Table 2. Comments on the choice of concentrations would be valuable.
Response 8: We have added some valuable information in Table 2. Please see Table 1. of the revised thesis.
Comments 9: Line 144: "The irradiance 300 mm from the surface of the sample is measured to be 50 W/m2." It is necessary to indicate the wavelength range of the UV radiation from the UV lamps.
Response 9: We have added the wavelength range of the UV radiation from the UV lamps: "It has a wavelength range of ultraviolet light from 300 to 400nm.", please see lines 147-148 of the revised thesis.
Comments 10: Figure 4, caption. The composition of the modified asphalt should be specified.
Response 10: We have specified the modified asphalt is SBS/CR modified. Please see Figure 3 of the revised thesis.
Comments 11: Line 231: “Fp of SBS&CR modified asphalt is the lowest among the four asphalts, indicating the lowest stiffness”. This is an incorrect statement. Fp is a measure of cohesive strength. Stiffness is the slope of the line from 0 to Fp.
Response 11: We have deleted the sentence.
Comments 12: Figure7a.The authors should use stress rather than force, i.e., the force divided by the cross-sectional area of the specimens.
Response 12: We have modified the force to tensile force, please see Figure 6a of the revised thesis.
Comments 13: Line 248: “rubber power modifier decreases the stiffness of the virgin asphalt, whereas the SBS and TiO2 modifiers increase the stiffness”. It's strength, not stiffness. The authors do not evaluate stiffness, which is the slope of the initial section of force-ductility curves.
Response 13: This expression has been revised and improved in the revised manuscript.Fp, Fy and Ff represent the tensile cohesive strength, yield strength and the final failure strength corresponds to the maximum ductility, respectively. Please see lines 243-244 of the revised thesis.
Comments 14: Line 277: “LTA aging is a more serious aging form than STA aging”. This is obvious since LTA aging consists of STA aging and then UV aging, if I understand correctly.
Response 14: We have deleted the sentence.
Comments 15: Line 409: “This founding may offer a microcosmic explanation.” It is unclear what the authors mean.Clarification is needed.
Response 15: This expression has been revised and improved in the revised manuscript. As follows: Thus, there is no doubt that the surface of SBS&CR modified asphalt is flatter than that of other asphalt, because it has the lowest surface roughness under the same aging condition. The smaller the difference between the phases, the less likely it is to produce phase stress concentration during the tensile process. This may be the micro reason for its good macro-scale tensile toughness. Please see lines 459-463 of the revised thesis.
Comments 16: Line 452: “it is found that the damage of virgin asphalt”. The type of damage should be clarified.Aging? Fracture due to rupture?
Response 16: We have deleted the sentence.
Comments 17: Lines 457, 459: “Fp, FY and Ff show”, “increase the W”. In the conclusion, it is better to use words rather than symbols because some people start reading the article with the conclusion.
Response 17: When the symbols first appeared, we added the corresponding explanation, Please see lines 483 -489 of the revised thesis.
With best regards
Round 3
Reviewer 3 Report
Comments and Suggestions for Authors
I still do not see the changes that the authors made to the article. The authors did not provide a manuscript with detailed corrections marked in color font or highlighted. In addition, I do not see the corrections that the authors made to the introduction of the article.
I still cannot recommend this article for publication for several reasons.
First, the article's introduction is weak and does not provide a comprehensive review of relevant studies on the research topic. The authors cite only 22 references to outdated papers. For example, the authors cite only three papers from 2022 and one paper from 2023, although there were tens of such papers in 2022-2024. In this way, the authors mislead the reader by claiming the uniqueness of their work.
Second, the authors do not compare their results on the effect of nanoparticles on the cohesive strength of bitumen with other works. There are many works in which the effect of nanoparticles of different natures (silica, clay, and so on) on the cohesion and adhesion of bitumen is investigated. The authors ignore these works. The presentation of experimental results without comparing them to previously obtained data is of little value and misleads the reader into thinking that the presented work is unique.
Third, the article's subject matter is not in line with the main aims and scopes of Polymers. The authors use a bitumen matrix, not a polymer matrix. Bitumen is not polymeric in nature, and the addition of SBS and crumb rubber does not turn it into a polymer. The authors could have strengthened the polymeric aspect of the article but did not do so.
Fourth, the article does not fit the theme of the special issue “Preparation, Structure and Characterization of Polymer/Cement Composites—3rd Edition.” It does not focus on obtaining polymer/cement composites and has no connection to cement materials (Portland cement, non-hydraulic slaked lime cement, etc.). The authors' addition of 2% titanium oxide to the bitumen does not turn it into cement.
For the reasons listed above, I cannot recommend this article for publication in Polymers.
I can recommend to the authors to strengthen the introduction to their article, compare the physical-mechanics of their samples with previously obtained bitumen nanocomposites, and submit the improved article to a journal having relevant aims and scopes, such as Materials, Buildings, or Applied Sciences.
Comments on the Quality of English LanguageThe English language requires moderate editing.